# Synthesis and Characteristics of Zn-Doped CuCrO$_2$ Transparent Conductive Thin Films

**Ruei-Sung Yu [1,2,\*] and Chen Chu [1]**

[1] Department of Photonics and Communication Engineering, Asia University, 500 Lioufeng Rd., Wufeng, Taichung 41354, Taiwan; lichking900@gmail.com

[2] Department of Medical Research, China Medical University Hospital, China Medical University, Taichung 40402, Taiwan

\* Correspondence: rsyu@asia.edu.tw; Tel.: +886-4-23323456; Fax: +886-4-23320718

**Abstract:** The effects of doping a p-type CuCrO$_2$ film with zinc on its structural and optoelectronic properties were investigated by experiments using CuCr$_{1-x}$Zn$_x$O$_2$ thin films ($x$ = 0, 0.025, 0.065, 0.085). An increase in the amount of zinc dopant in the thin films affected the lattice constant and increased its Gibbs free energy of phase transformation. Cross-sectional images of the CuCrO$_2$ thin film samples exhibited a dense polygonal microstructure and a surface morphology with protruding nanoscale granules. With the increase in the amount of Zn dopant, the surface roughness decreased, thereby increasing the amount of incident photons as well as the visible-light transmittance and ultraviolet-light absorption of the thin films. With the zinc doping in the CuCrO$_2$ thin films, the band gap increased from 3.09 to 3.11 eV. The substitution of Cr$^{3+}$ with Zn$^{2+}$ forms hole carriers in the crystals, which was demonstrated by X-ray photoelectron spectroscopy and Hall effect measurements. The conductivities and carrier concentrations of the Zn-doped CuCrO$_2$ thin films were greater than those of undoped CuCrO$_2$. The CuCr$_{1-x}$Zn$_x$O$_2$ film ($x$ = 0.065) exhibited the best optoelectronic properties; its carrier concentration and resistivity were $1.88 \times 10^{17}$ cm$^{-3}$ and 3.82 $\Omega$cm, respectively.

**Keywords:** CuCrO$_2$; zinc-doped; thin film; structure; optoelectronic properties

## 1. Introduction

Transparent conductive oxides (TCOs), with a band gap of approximately 3.0 eV, exhibit visible-light transmittance, UV-light absorbance, and good conductivity. Because of these interesting characteristics, TCOs can meet the requirements for transparent, translucent p–n components [1–3]. Recently, studies of the application of p-type materials in p–n junctions have reportedly converted n-type TCOs into p-type materials. For instance, novel p–n diodes could be developed by converting n-type ZnO to its p-type form [4,5].

P-type TCO delafossite CuCrO$_2$ exhibits optoelectronic, catalytic, and magnetic properties, as well as potential for development and applications [6,7]. Because thin-film materials are crucial components of various products, CuCrO$_2$ thin films were focused on in this study. It is anticipated that the CuCrO$_2$ thin film possesses optimal physical properties, which is conducive to the development of practical devices. The conductive properties of CuCrO$_2$ can be improved by the addition of doping elements to increase its hole concentration and reduce its resistance. A majority of studies have reported the doping of CuCrO$_2$ with Mg$^{2+}$ [8–11]. On the other hand, in this study, zinc (Zn$^{2+}$) was doped in CuCrO$_2$. Zinc was used as the doping element because it is a non-toxic, non-precious metallic element. The substitution of Cr$^{3+}$ at lattice positions with Zn$^{2+}$ forms point defects, which generate hole carriers; hole carriers in turn increase conductivity. Previous studies have applied Zn-doping of CuCrO$_2$ to nanoparticles and thin films [12–14]. In contrast to these relevant studies, different processing

conditions were employed, that is, a lower phase-transformation temperature, lower partial oxygen pressure, higher passive annealing atmosphere of argon, and more varied quantities of the Zn dopant. This experiment aimed to investigate the effects of zinc doping on the structural, optical, and electrical properties of $CuCrO_2$. We have confirmed that the zinc dopant in $CuCrO_2$ thin film can increase the transmittance and carrier concentration and reduce the resistivity. This research demonstrates effective doping and establishes a theory of zinc doping on the structural and optoelectronic characteristics of $CuCrO_2$.

## 2. Materials and Methods

The sol–gel method was employed to prepare p-type semiconductor thin films. The substrate was quartz glass. Before modulation, the precursor solutions used were copper acetate (Showa, Saitama, Japan, 98%), chromium acetate (Showa, Saitama, Japan, 98%), zinc acetate (Showa, 99%), triethanolamine (Tedia, Fairfield, OH, USA, 99%), and anhydrous ethanol (Shimakyu, Niigata, Japan, 99.5%). The $CuCr_{1-x}Zn_xO_2$ films with different Cr/Zn ratios ($x = 0, 0.025, 0.065, 0.085$) were prepared, and various experimental parameters were used to modulate the proportions for the experiment. The amount of copper was 0.02 mol; the amounts of chromium were 0.02 mol ($x = 0$), 0.0195 mol ($x = 0.025$), 0.0187 mol ($x = 0.065$), and 0.0183 mol ($x = 0.085$); and the amounts of zinc were 0 mol ($x = 0$), 0.0005 mol ($x = 0.025$), 0.0013 mol ($x = 0.065$), and 0.0017 mol ($x = 0.085$). Anhydrous ethanol (99.5%) was added, and then the mixture was stirred at a rotational speed of 200 rpm for 2 h at room temperature. Triethanolamine was added, and then the mixture was stirred for 24 h to complete the modification of the precursors. The deposition of thin films was carried out by spin coating at a spinning speed of 1500 rpm for 15 s. After completion, the thin films were subjected to a pre-annealing treatment: the films were dried at 300 °C for 10 min, and the temperature was then increased to 500 °C for 2 h. This step was performed three times to complete the spin coating and pre-annealing treatment. Finally, the film specimens were placed in a tubular atmosphere furnace filled with argon gas (99.99% purity) at a flow rate of 200 sccm. The partial pressure of oxygen in the tube was very low. The temperature was 600 °C for 2 h to carry out controlled-atmosphere argon annealing.

An X-ray diffraction (XRD, D8 Discover, Bruker, Karlsruhe, Germany) measurement was employed to analyze the crystal structures of the $CuCr_{1-x}Zn_xO_2$ thin films ($x = 0, 0.025, 0.065, 0.085$). The XRD was operated at 40 KV, 30 mA with Cu K$\alpha$ radiation (wavelength = 1.54 Å). The cross-sectional microstructure and thickness of films were analyzed by field emission scanning electron microscopy (FESEM, JSM-7800F, JEOL, Tokyo, Japan). Atomic force microscopy (AFM, Dimension Icon, Bruker, Karlsruhe, Germany) was performed to measure the surface morphology and roughness of the film samples. The electron binding energy of the undoped and Zn-doped $CuCrO_2$ was analyzed using X-ray photoelectron spectroscopy (XPS, K-Alpha, Thermo Scientific, Loughborough, UK). The XPS instrument was operated at a wavelength of 0.83 nm with Al target radiation. Thin-film transmittance measurements were conducted on an ultraviolet–visible spectroscopy system. The Hall effect measurement was employed to analyze the electrical properties of the $CuCr_{1-x}Zn_xO_2$ films. Hall effect measurement can identify the electrical features of resistivity, mobility, concentration, and Hall coefficient. The measured parameter of the magnetic field was 0.6 T at room temperature.

## 3. Results and Discussion

Figure 1 shows the XRD results for each film specimen. All of the undoped and Zn-doped $CuCrO_2$ thin films had delafossite structures with (006), (101), (012), (104), (018), (110), and (00<u>12</u>) diffraction peaks. Zn and Zn-related oxide structures were absent because the solid solution of Zn was in the $CuCrO_2$ lattice, and the doping concentration did not exceed the solubility. However, traces of CuO and $CuCr_2O_4$ remained in the $CuCr_{1-x}Zn_xO_2$ ($x = 0.085$) thin film, related to the fact that CuO and $CuCr_2O_4$ are at low-temperature stable phases [15]. The phase transformation of $CuCrO_2$ is as follows:

$$CuO + CuCr_2O_4 \rightarrow 2CuCrO_2 + \tfrac{1}{2}O_2 \tag{1}$$

The thermodynamics for this phase transformation may be described as follows [16]:

$$\Delta G = \Delta G° + RT \ln(pO_2) \tag{2}$$

$$\Delta G = -RT \ln(pO_2)_{equil.} + RT \ln(pO_2) \tag{3}$$

$$\Delta G = RT \ln(pO_2/(pO_2)_{equil.}) \tag{4}$$

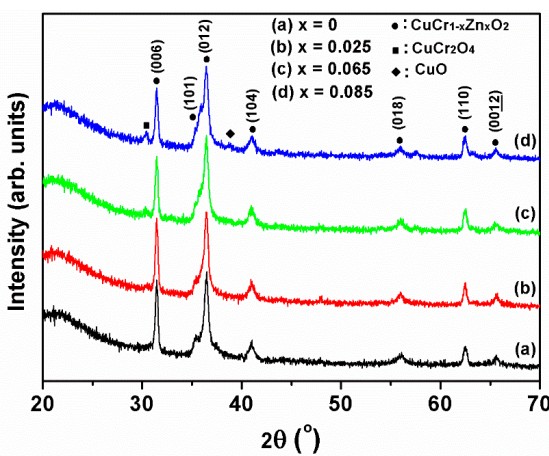

**Figure 1.** X-ray diffraction patterns of the undoped and Zn-doped $CuCrO_2$ thin films.

Here, $\Delta G$ is the transfer of Gibbs free energy (J/mol), $\Delta G$ is the change in the standard Gibbs free energy (J/mol), $R$ is the gas constant, $T$ is the absolute temperature, $pO_2$ is the partial pressure of oxygen in the flowing gas, and $(pO_2)_{equil.}$ represents the specific partial pressure of oxygen, with $CuO$, $CuCr_2O_4$, and $CuCrO_2$ in the material at thermodynamic equilibrium. According to the thermodynamic data, the changes in the Gibbs free energy may be explained by the temperature and partial pressure of oxygen [17]. According to theories in materials science, $Cr^{3+}$ can be substituted with $Zn^{2+}$ as the ionic radii of $Cr^{3+}$ and $Zn^{2+}$ are 0.63 and 0.74 Å, respectively (discussed in a later section with the band gap, XPS and Hall effect measurements). Using the total pattern analysis solution to determine the lattice constant of $CuCr_{1-x}Zn_xO_2$, the a- and c-axis constants increased with the Zn dopant content (Table 1). XRD JCPDS data file No. 89-6744 indicates that the delafossite structure lattice constants of a and c are 2.973 and 17.100 Å, respectively. This difference in dimensions of the ions distorts the lattice and forms a stress field [18]. The greater stress field in $CuCr_{1-x}Zn_xO_2$ ($x = 0.085$) caused by the amount of zinc in the solid solution impeded atomic diffusion during $CuCrO_2$ phase transformation, increasing its required Gibbs free energy. Under the same annealing conditions (600 °C), residual low-temperature structures of $CuO$ and $CuCr_2O_4$ were observed in the thin film. According to the thermodynamic theory, the $CuO$ and $CuCr_2O_4$ phases may be removed by increasing the annealing temperature or by decreasing the partial pressure of oxygen (both were not performed in our study). Lowering the Gibbs free energy permits the reaction between $CuO$ and $CuCr_2O_4$, affording $CuCrO_2$. A previous related study has used Equations (2)–(4) for its thermodynamic calculations and for the construction of $CuCrO_2$, $CuO$, and $CuCr_2O_4$ phase diagrams describing the relationship between the partial pressure of oxygen and annealing temperature [17]. Several arguments can be formulated by this study and the previous study: 1. The appropriate chemical composition, low partial pressure of oxygen, and high temperature are crucial conditions for the formation of single-phase $CuCrO_2$; 2. $CuO$ and $CuCr_2O_4$ react to form $CuCrO_2$ at a high process temperature [15]; 3. An oxygen partial pressure of less than $10^{-3}$ atm and a temperature of ≥600 °C enable the formation of a single-phase $CuCrO_2$ structure.

**Table 1.** Lattice constants of delafossite $CuCr_{1-x}Zn_xO_2$ thin films ($x = 0$, 0.025, 0.065 and 0.085 Å).

| $CuCr_{1-x}Zn_xO_2$ Thin Films | $a$-Axis | $c$-Axis |
|:---:|:---:|:---:|
| $x = 0$ | 2.972 | 17.069 |
| $x = 0.025$ | 2.974 | 17.070 |
| $x = 0.065$ | 2.974 | 17.074 |
| $x = 0.085$ | 2.975 | 17.071 |

FESEM cross-sectional images of the thin films revealed irregularly shaped, densely packed microstructures. The formation of these microstructures was dependent on the annealing process. Zn doping did not significantly affect this structural characteristic, as evidenced by the results for undoped and heavily doped $CuCr_{1-x}Zn_xO_2$ ($x = 0$ and 0.085; Figure 2a,b). The thicknesses of the $CuCr_{1-x}Zn_xO_2$ films ($x = 0$, 0.025, 0.065, 0.085) were in the range of 163 to 169 nm. The minor variation of the film thickness was in agreement with the processing capabilities of sol–gel spin coating and annealing treatment, and the cross-sectional microstructures were similar to those reported in the literature [19,20].

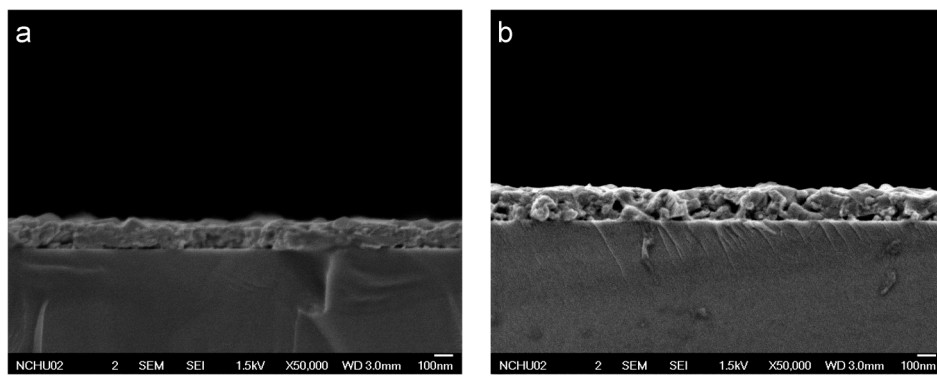

**Figure 2.** Images of the cross-sectional microstructures of the $CuCr_{1-x}Zn_xO_2$, where $x = $ (**a**) 0 and (**b**) 0.085 of the undoped and heavily doped thin films.

As can be observed by the two- and three-dimensional AFM images in Figures 3a and 4a, the surface of the undoped $CuCrO_2$ thin film consisted of uniformly dispersed granules of different sizes. The root-mean-square roughness of the thin film surface was 20.6 nm. Figures 3b–d and 4b–d show the two- and three-dimensional AFM surface morphologies of Zn-doped $CuCrO_2$, which were similar to the surface structures of undoped $CuCrO_2$. The surface root-mean-square roughness values of the $CuCr_{1-x}Zn_xO_2$ thin films with $x$ values of 0.025, 0.065, and 0.085 were 19.7, 15.4, and 15.8 nm, respectively. Therefore, the increase in the Zn dopant content resulted in a decrease in the surface root-mean-square roughness. This decrease is related to the stress field resulting from the insertion of zinc into the delafossite lattice. During annealing, the stress field hinders the migration of atoms, dislocations, and grain boundaries during growth, thereby limiting the microstructural growth of surface granules and decreasing the root-mean-square roughness of the surface.

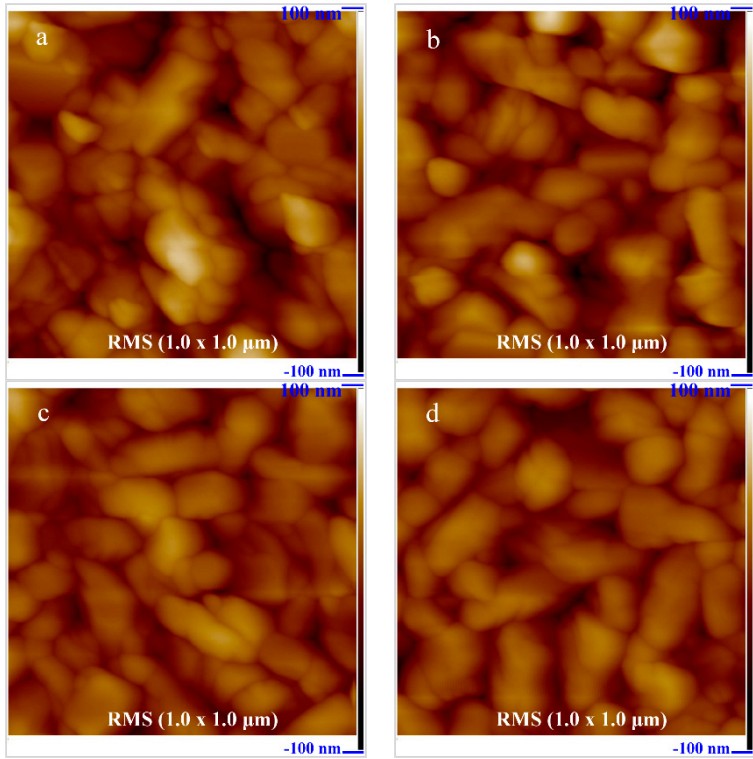

**Figure 3.** Atomic force microscopy (AFM) two-dimensional (2D) images of the surface morphologies of the $CuCr_{1-x}Zn_xO_2$ films for $x =$ (**a**) 0, (**b**) 0.025, (**c**) 0.065, and (**d**) 0.085.

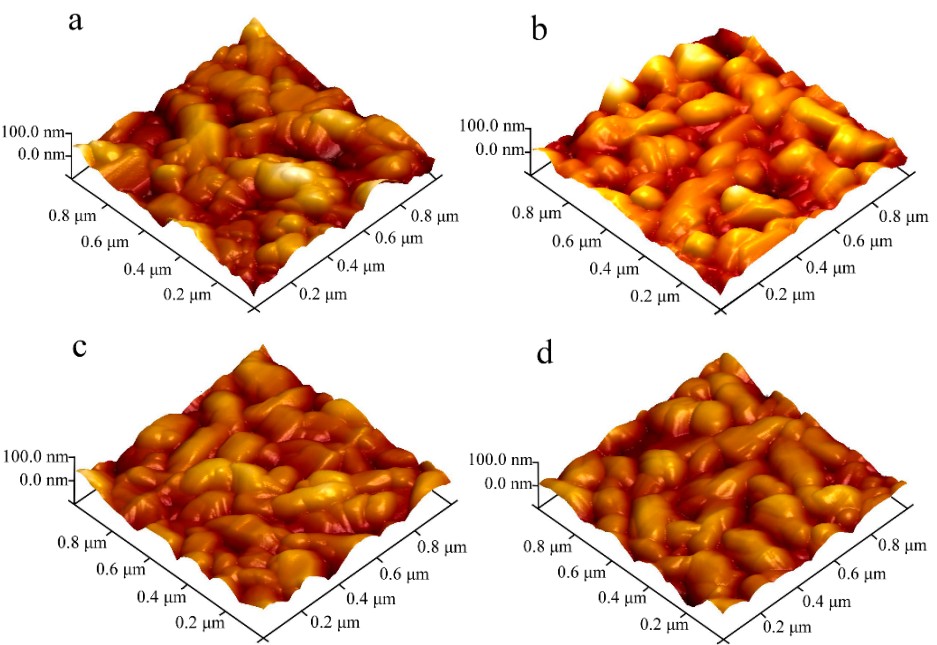

**Figure 4.** The three-dimensional (3D) surface images of the AFM measurements for the $CuCr_{1-x}Zn_xO_2$, where x = (**a**) 0, (**b**) 0.025, (**c**) 0.065, and (**d**) 0.085.

Transmittance measurements (Figure 5) indicated that all of the thin films exhibit translucency at wavelengths longer than around 380 nm. The transmittance spectra of the films were similar, and the transmittance of the undoped $CuCrO_2$ thin film was slightly lower. The fundamental transmittance edge of the Zn-doped thin films exhibited a blue-shift phenomenon. With increasing zinc content, the transmittance of the $CuCr_{1-x}Zn_xO_2$ thin films increased. At wavelengths between 500 and 800 nm, the

transmittance of the Zn-doped CuCrO$_2$ thin film increased by approximately 2% to 7%. This increase is related to the lowering of surface roughness, reducing the diffuse reflection of photons; the normal directions of the formations were different while the incident parallel light ray impinged a rough surface. Therefore, a lower surface roughness of the film causes a low diffuse reflection and an increase in the incident light ray (photons), resulting in increased transmittance. At a wavelength of 800 nm, the transmittances of the undoped and Zn-doped CuCr$_{1-x}$Zn$_x$O$_2$ thin films were approximately 68% and 70–75%, respectively.

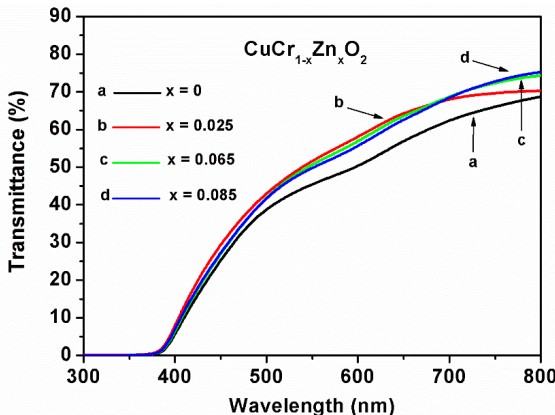

**Figure 5.** Transmittances of the CuCr$_{1-x}$Zn$_x$O$_2$ thin films.

Initially, the absorption coefficients of all of the CuCrO$_2$ thin films between wavelengths of 400 and 800 nm (Figure 6) were relatively small, considerably increasing at wavelengths shorter than 400 nm. The thin films mainly absorbed ultraviolet light with short wavelengths; thus, doping the CuCr$_{1-x}$Zn$_x$O$_2$ films ($x$ = 0.065, 0.085) with higher amounts of Zn can increase the absorption of photons with wavelengths of 300 to 350 nm. This trend is related to the low surface roughness and high carrier concentration. Factors affecting the absorption coefficients of CuCrO$_2$ thin films include (1) the surface roughness, which affects the amount of incident rays (photons) into the thin film; (2) typically, the increase in the dopant content results in different lattice arrangement symmetries, thus changing the wave and path of ray (photons) propagation and resulting in increased scattering (the degree of lattice distortion caused by the dopant content of the thin films affects the optical path of rays inside the thin films); (3) the level of carrier concentration in the material can affect the amount of photons absorbed. The unit volume of film has a high density of carriers; this also results in a high probability and number of absorbed photons. In summary, thin films with high absorption coefficients exhibit characteristics of low roughness, a low degree of diffuse reflection, high incident photons, and a high probability of photons being absorbed.

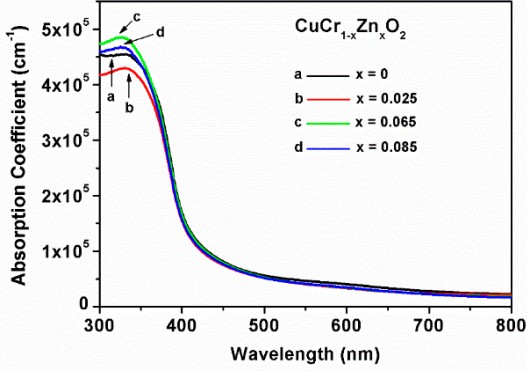

**Figure 6.** Absorption coefficients of the CuCr$_{1-x}$Zn$_x$O$_2$ films.

Tauc's relation of $(\alpha h\nu)^2$ versus h$\nu$ was plotted to measure the direct band gap of the material [21]. The band gap of undoped $CuCrO_2$ film was 3.09 eV (Figure 7a). As shown in Figure 7b–d, the band gaps of the $CuCr_{1-x}Zn_xO_2$ films with $x$ values of 0.025, 0.065, and 0.085 were 3.09, 3.10, and 3.11 eV, respectively. Therefore, doping the $CuCrO_2$ thin film with zinc leads to an increased band gap. The trend is similar to the blue-shift phenomenon as observed in the transmittance measurements. As the energy level of Cu $3d^{10}$ in the delafossite oxide structure is close to the O 2$p$ levels, the hybridized orbitals can cause the delocalization of the hole states, conferring mobility to the holes [22]. With the disruption of the interatomic bonds adjacent to the holes by external energy, valence electrons fill the positions of the holes, resulting in hole displacement. In the zinc-doped $CuCrO_2$ film, $Zn^{2+}$ replaces $Cr^{3+}$; concurrently, Zn 4$s$ forms an acceptor energy level slightly above the high valence band in the band gap structure. Energy at room temperature can excite electronic transitions from the valence band to these acceptor energy levels, causing an increase in hole carrier concentration in the valence band and an increase in the band gap of the $CuCrO_2$ thin films. The band gap values obtained in this study (3.09–3.11 eV) are similar to those reported for $CuCrO_2$ subjected to extrinsic doping [13,23,24].

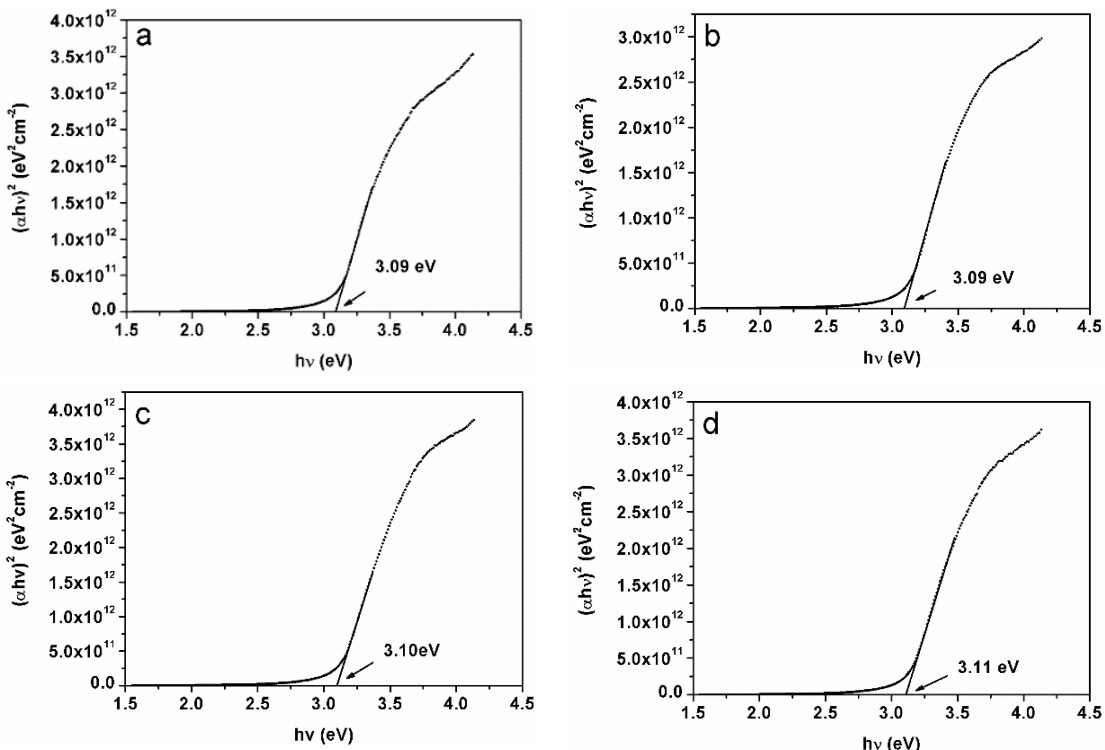

**Figure 7.** Direct band gaps of the $CuCr_{1-x}Zn_xO_2$ films for x = (**a**) 0, (**b**) 0.025, (**c**) 0.065, and (**d**) 0.085.

Using the mathematical theory of extrapolation of measurement data, the calculation can be performed in line with the actual measurement results. If the energy gap for $CuCr_{1-x}Zn_xO_2$ ($x = 0.085$) is assumed to be unknown, then the known data for $CuCr_{1-x}Zn_xO_2$ ($x = 0, 0.025, 0.065$) can be used to calculate the band gap by extrapolation. The calculated band gap of $CuCr_{1-x}Zn_xO_2$ ($x = 0.085$) was 3.105 eV. This value is close to the measured energy gap (3.11 eV).

Figure 8a–d shows the results of XPS measurements of the $CuCr_{1-x}Zn_xO_2$ thin films ($x = 0, 0.025, 0.065, 0.085$). The photoelectron binding energies of Cu 2$p_{3/2}$, Cu 2$p_{1/2}$, Cr 2$p_{3/2}$, Cr 2$p_{1/2}$, and O1$s$ were 932.7, 952.5, 576.8, 586.4 and 530.5 eV, respectively; their corresponding atomic valences were monovalent cation (Cu$^+$), trivalent cation (Cr$^{3+}$), and divalent anion (O$^{2-}$), respectively. These values of electron binding energy are close to those reported in the literature [25]. The binding energies of Zn 2$p_{3/2}$ were 1021.7 eV, indicating that zinc was a divalent cation (Zn$^{2+}$). Based on the theories of ceramic science, since the $CuCr_{1-x}Zn_xO_2$ thin films are crystalline delafossite, the anions and cations

bond to one another. The increase of carrier concentration by zinc doping observed in the Hall effect measurement suggests that zinc replaces chromium site (see below).

XPS is sensitive to surface contamination [26]. Before conducting the XPS analysis, the surfaces of the $CuCr_{1-x}Zn_xO_2$ were bombarded with argon ions to remove absorbed oxygen elements and contaminants, in order to acquire the correct compositions. The stoichiometric compositions of $CuCr_{1-x}Zn_xO_2$ are $Cu:Cr_{1-x}Zn_x:O = 25:25:50$. For each set of quantitative values of Cu, Cr, Zn, and O in these films, the anion O contents were 57.57–58.56%, whereas the cation Cu and $Cr_{1-x}Zn_x$ elements were less than the stoichiometric ratio inside the films. The Cu contents were 17.66–19.20%. The Cr contents were 22.95–23.78%. The Zn contents were 0.18–0.44%. The contents of the O were twice as high as that of the Cu and $Cr_{1-x}Zn_x$ elements. This is favorable for the generation point defects of hole carriers, such as Cu and Cr vacancies or interstitial oxygen. In this study, the Zn-doping technique was also implemented to enhance hole carrier concentrations. The $CuCr_{1-x}Zn_xO_2$ ($x = 0, 0.025, 0.065$) films with different Zn-doping contents displayed the same single-phase structure, as demonstrated by XRD. No impurity phase contributions to the sub-peak were observed in the $CuCr_{1-x}Zn_xO_2$ film ($x = 0$, 0.025, 0.065). The $CuCr_{1-x}Zn_xO_2$ ($x = 0.085$) film had traces of CuO and $CuCr_2O_4$, and the impurity phases may contribute to the sub-peaks in the spectra. The Zn element is present in the film, which suggests the effect of the O–Zn bond in a minor peak shift of oxygen. No Cu satellite transition was observed in the diagram.

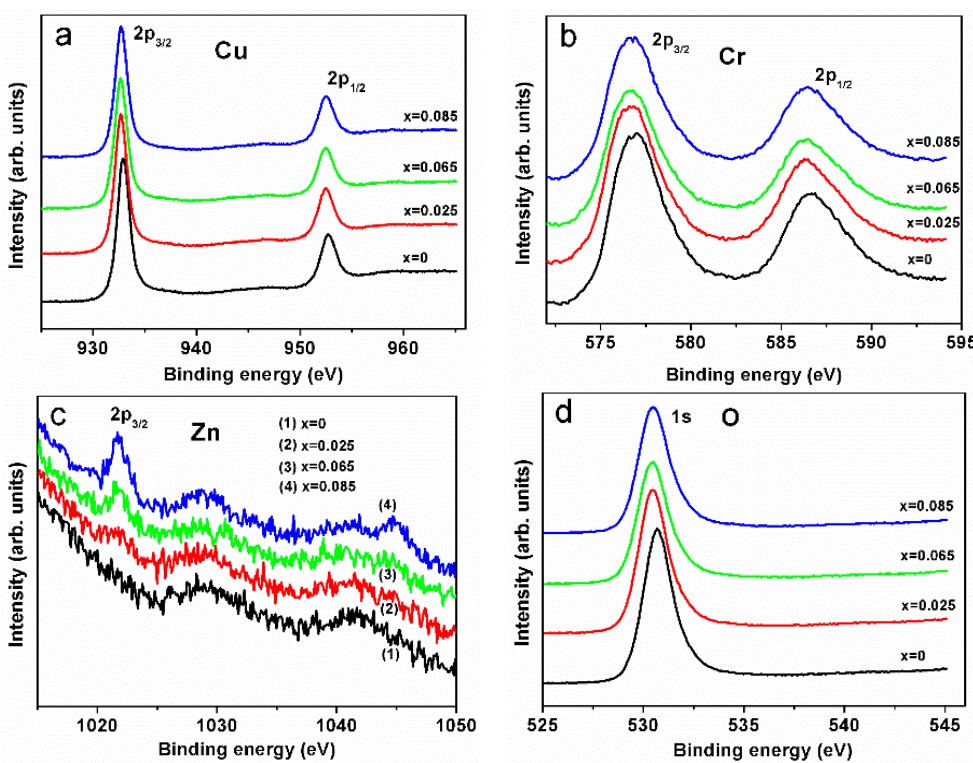

**Figure 8.** The photoelectron spectra of (**a**) Cu, (**b**) Cr, (**c**) Zn and (**d**) O of the $CuCr_{1-x}Zn_xO_2$ thin films ($x = 0, 0.025, 0.065, 0.085$).

Table 2 shows the relationship between the Zn contents and optoelectronic properties of the $CuCr_{1-x}Zn_xO_2$ films. The Hall effect measurements for the electrical properties of $CuCr_{1-x}Zn_xO_2$ thin films ($x = 0, 0.025, 0.065, 0.085$) are as follows: for the undoped $CuCr_{1-x}Zn_xO_2$ ($x= 0$) thin film, the resistivity was 34.35 $\Omega$cm, and the carrier concentration was $1.09 \times 10^{15}$ cm$^{-3}$. The substitution of $Cr^{3+}$ with $Zn^{2+}$ in the solid solution permitted the lattice point defects to generate hole carriers. The effect of the $Cr^{3+}$–$Zn^{2+}$ substitution on the carrier formed in delafossite-$CuCrO_2$ may be described as follows:

$$(Cu_{Cu})^\times + (Cr_{Cr})^\times + 2(O_O)^\times \rightarrow (Cu_{Cu})^\times + (Cr_{Cr})^\times + (Zn_{Cr})' + 2(O_O)^\times + h^\cdot \qquad (5)$$

Here, $(Cu_{Cu})^\times$, $(Cr_{Cr})^\times$, and $(O_O)^\times$ represent copper, chromium and oxygen in the original lattice sites, respectively; $(Zn_{Cr})'$ represents that $Zn^{2+}$ replaces $Cr^{3+}$, which then creates a hole ($h^\cdot$) carrier. Hence, the carrier concentrations in $CuCr_{1-x}Zn_xO_2$ with $x = 0.025$, 0.065 and 0.085 after doping of the $CuCrO_2$ thin films with zinc were $8.80 \times 10^{15}$, $1.88 \times 10^{17}$ and $1.67 \times 10^{17}$ cm$^{-3}$, respectively. Their corresponding resistivity values were 20.16, 3.82 and 3.98 $\Omega$cm. These resistivities are in a lower range of $1.16 \times 10^2$ to $8.57 \times 10^4$ $\Omega$cm, as reported in our previous study on intrinsic $CuCrO_2$ films prepared by the sol–gel method [19]. In addition, the electrical properties are similar to those reported for Zn-doped $CuCrO_2$ (2.12–3.84 $\Omega$cm) [13]. The $CuCr_{1-x}Zn_xO_2$ ($x = 0.065$, 0.085) films with lower resistivities were 3.82 and 3.98 $\Omega$cm, respectively, and the carrier mobilities were 8.65 and 9.38 cm$^2$/Vs, respectively. The values are close to conventional p-type semiconductors. General speaking, when compared to n-type semiconductors, p-type materials have lower mobility due to the fact that a hole current can only be evolved when electron holes are filled with valence electrons in the neighborhood, not free electrons. Positive Hall coefficients indicate that the material is a p-type semiconductor. The results thus confirm that the zinc dopant in the $CuCrO_2$ thin film increases its carrier concentration and reduces its resistivity. On the contrary, if Zn-dopant atoms are at interstitial sites ($Zn_i^{2+}$) or substitute copper sites ($Cu^+$), then they contribute to the outer orbital electrons. The combination of an electron with a hole results in a lower hole carrier concentration, thereby increasing the resistivity. This study therefore demonstrated the effective doping of p-type $CuCrO_2$ with $Zn^{2+}$, which substitutes $Cr^{3+}$.

**Table 2.** $CuCr_{1-x}Zn_xO_2$ thin films' Zn content versus semiconductor characteristics.

| Specimen | Zn Content (at %) | Band Gap (eV) | Resistivity ($\Omega$cm) | Carrier Concentration (cm$^{-3}$) |
|---|---|---|---|---|
| $CuCrO_2$ | – | 3.09 | 34.35 | $1.09 \times 10^{15}$ |
| $CuCr_{0.975}Zn_{0.025}O_2$ | 0.18 | 3.09 | 20.16 | $8.80 \times 10^{15}$ |
| $CuCr_{0.935}Zn_{0.065}O_2$ | 0.26 | 3.10 | 3.82 | $1.88 \times 10^{17}$ |
| $CuCr_{0.915}Zn_{0.085}O_2$ | 0.44 | 3.11 | 3.98 | $1.67 \times 10^{17}$ |

Previous literature has reported the importance of developing p-type transparent/oxide semiconductors and its applicability to device electronics [27,28]. The work functions of conventional n/p-type TCOs and their problems and solutions have also been discussed comprehensively. We share the same purposes, which is conducive to the practical optoelectronic applications of these semiconductors. Fabrication of such a p–n heterojunction device makes it possible to develop the p-type $CuCr_{1-x}Zn_xO_2$ applications, such as selective contacts in photovoltaics, and p-channel transparent film transistors. To achieve the available temperature of the mass-production below 400 °C, the high energy generated by the excimer laser is applied to the surface of the $CuCrO_2$ film, and a thermal energy effect is generated only at a depth of 100 nm on the film surface. Therefore, not too much thermal energy is transferred to the substrate or material underneath. This annealing method is beneficial for future $CuCrO_2$ development.

## 4. Conclusions

In this study, the sol–gel method, spin-coating method, and annealing treatment were employed to prepare Zn-doped $CuCrO_2$ thin films. Zn in the solid solution can exchange with Cr at the lattice sites, which changes the structure and optoelectronic characteristics of the thin film. All of the $CuCr_{1-x}Zn_xO_2$ thin films ($x = 0$, 0.025, 0.065) exhibited a single-phase structure; at higher amounts of the Zn dopant in $CuCr_{1-x}Zn_xO_2$ ($x = 0.085$), low-temperature phase structures of $CuO$ and $CuCr_2O_4$ were retained. The film thicknesses ranged from 163 to 169 nm. Doping zinc into the $CuCrO_2$ thin film gradually reduced the surface roughness from 20.6 to 15.4 nm, thereby increasing the visible-light transmittance and capability of UV-light absorption. The Zn-doped $CuCrO_2$ thin film exhibited an increased hole concentration from $10^{15}$ to $10^{17}$ cm$^{-3}$ as well as an increase in its band gap from 3.09 to 3.11 eV. For intrinsic and optimal doping of $CuCr_{1-x}Zn_xO_2$ ($x = 0$, 0.065), resistivities of the thin films were

respectively 34.35 and 3.82 $\Omega$cm. The substitution of $Cr^{3+}$ with $Zn^{2+}$ in the $CuCr_{1-x}Zn_xO_2$ thin film was identified by the XPS and Hall effect measurements. Doping $CuCrO_2$ with Zn is therefore suitable for improving its electrical and optical properties. The $CuCr_{1-x}Zn_xO_2$ film could be a candidate for the p-type semiconductor layer in practical devices.

**Author Contributions:** Y.R.-S. contributed in the conceptualization, methodology, formal analysis, writing-original draft preparation, writing-review and editing, supervision, project administration, and funding acquisition. C.C. contributed in the software, investigation, validation, resources, data curation, and visualization. All authors approved the final version of the paper to be submitted.

**Funding:** This research was funded by the Ministry of Science and Technology of Taiwan (No. MOST 101-2221-E-468-007-MY2).

**Acknowledgments:** We would like to acknowledge Michael Burton of Asia University.

**Conflicts of Interest:** The authors declare no conflict of interest.

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
