# Peer review of "Synthesis and Characteristics of Zn-Doped CuCrO2 Transparent Conductive Thin Films"

_coatings, doi:10.3390/coatings9050321_

Reviewer 1 Report

1. Line 53: what is the substrate?

2. Line 71/76: wavelength and/or energy of employed x-ray sources for XRD and XPS have to be mentioned.

3. Line 72: was the stoichiometry of the deposited films assumed or independently confirmed? How?

4. Line 77: a few more details reagarding the Hall measurements should be provided.

5. Table 1: provide literature values of a and c (where available). I would prefer the use of nm instead of Angstroem.

6. Fig. 7 and text: the band gap differences are quite small. Is it significant and what is the accuracy of the band gap determination?

7. Fig. 8: There are a few points which should be further considered/analysed and addressed in the text. It is common that a peak fitting is performed. For example, are these single peaks or composed of 2 or more peaks. For example, Cr and O are asymmetric which suggests that these are composed of at least 2 sub-peaks. What is the origin of these sub-peaks? Do you observed Cu satellite transitions? Further information can be found in, e.g., Majumdar et al., Coatings 7, 64 (2017). Can you derive the surface stoichiometry and does it differ from the bulk stoichiometry?     

8. Line 226: Hall measurement results should be given in a table.

Author Response

Thank you for the reviewers’ positive comments. I have carefully read the comments and revised the manuscript according to reviewers’ suggestions. All the changes of technical content are marked blue font in the revised manuscript. A brief description of my response to each point raised by the reviewers is as follows.

#Reviewer 1

Comments and Suggestions for Authors

1. Line 53: what is the substrate?

The substrate was quartz glass.

2. Line 71/76: wavelength and/or energy of employed x-ray sources for XRD and XPS have to be mentioned.

The XRD was operated at 40 KV, 30mA with CuKα radiation (wavelength=1.54Å).

The XPS instrument was operated at wavelength of 0.83 nm with Al target radiation.

3. Line 72: was the stoichiometry of the deposited films assumed or independently confirmed? How?

XPS is sensitive to surface contamination [26]. Before conducting the XPS analysis, the surfaces of the CuCr1−xZnxO2 were bombarded with argon ions to remove absorbed oxygen elements and contaminants, in order to acquire the correct compositions. The stoichiometric compositions of CuCr1−xZnxO2 are Cu: Cr1−xZnx:O = 25:25:50. For each set of quantitative values of Cu, Cr, Zn, and O in these films, the anion O contents were 57.57 at% – 58.56 at%; whereas the cation Cu and Cr1−xZnx elements were less than the stoichiometric ratio inside the films. The Cu contents were 17.66 at% –19.20 at%. The Cr contents were 22.95 at%–23.78 at%. The Zn contents were 0.18at% –0.44 at%. The contents of the O were high twice that of the Cu and Cr1−xZnx elements. It is favorable for the generation point defects of hole carriers, such as Cu and Cr vacancies or interstitial oxygen. In this study, the Zn doping technique was also implemented to enhance hole carrier concentrations.

26.   A. Majumdar; S. Drache; H. Wulff; A. K. Mukhopadhyay; S. Bhattacharyya; C. A. Helm; R. Hippler, Strain effects by surface oxidation of Cu3N thin films deposited by DC magnetron sputtering, Coatings 2017, 7, 64; doi: 10.3390 /coatings7050064

4. Line 77: a few more details reagarding the Hall measurements should be provided.

Hall effect measurement can identify the electrical features of resistivity, mobility, concentration, and Hall coefficient. The measured parameter of magnetic field was 0.6 T at room temperature.

5. Table 1: provide literature values of a and c (where available). I would prefer the use of nm instead of Angstroem.

XRD JCPDS data file No. 89-6744 indicates that the delafossite structure lattice constants of a and c are 2.973 and 17.100 Ã…, respectively.

Thank you for your comments.

6. Fig. 7 and text: the band gap differences are quite small. Is it significant and what is the accuracy of the band gap determination?

We checked the band gap determination carefully. These data are correct.

7. Fig. 8: There are a few points which should be further considered/analysed and addressed in the text. It is common that a peak fitting is performed. For example, are these single peaks or composed of 2 or more peaks. For example, Cr and O are asymmetric which suggests that these are composed of at least 2 sub-peaks. What is the origin of these sub-peaks? Do you observed Cu satellite transitions? Further information can be found in, e.g., Majumdar et al., Coatings 7, 64 (2017). Can you derive the surface stoichiometry and does it differ from the bulk stoichiometry?  

The CuCr1−xZnxO2 (x = 0, 0.025, 0.065) films with different Zn doping contents displayed the same single-phase structure, as demonstrated by XRD. No impurity phase contribute to the sub-peak were observed in the CuCr1−xZnxO2 film (x = 0, 0.025, 0.065). The CuCr1−xZnxO2 (x =0.085) film with traces of CuO and CuCr2O4, which the impurity phases may contribute sub-peak in the spectra. The Zn element is present in the film, which suggests the effect of O-Zn bond in a minor peak shift of oxygen. No Cu satellite transition was observed in the diagram.

Before conducting the XPS, the surfaces of the films were bombarded with argon ions to remove absorbed oxygen elements and contaminants, we in order to obtain the correct film inside compositions.

8. Line 226: Hall measurement results should be given in a table.

Table 2 shows the relationship between the Zn contents and optoelectronic properties of the CuCr1−xZnxO2 films.

Table 2 CuCr1−xZnxO2 thin films’ Zn content versus semiconductor characteristics.

Specimens

Mg

content

(at%)

Band

gap

(eV)

Resistivity (Ωcm)

Carrier concentration (cm-3)

CuCrO2

–

3.09

34.35

1.09 ×1015

CuCr0.975Zn0.025O2

0.18

3.09

20.16

8.80 ×1015

CuCr0.935Zn0.065O2

0.26

3.10

3.82

1.88 ×1017

CuCr0.915Zn0.085O2

0.44

3.11

3.98

1.67 ×1017

Reviewer 2 Report

In this paper, authors have successfully fabricated Zn doped CuCrO2 thin films for p-type TCO using solution-process. This paper is scientifically interesting for a variety of applications for optoelectronics devices. Therefore, this paper should be published in this journal. However, before this work is publishable, the authors should respond to the following concerns.

1.I agree with the importance of developing p-type TCOs. I suggest several papers to enhance the motivation of this work: J. Kim et al., Advanced Materials Interfaces, 5(23), 1801307., T. Jun et al., Advanced Materials 30 (12), 1706573.

It will be better that authors can explain the detailed necessity of p-type TCOs in the introduction part of this paper. Work functions of conventional n-type TCOs and their problems can be discussed.   

2. If possible, authors should comment on the work function of the CuCrO2.

3.Even if the mobility is low, authors must indicates the Hall mobility of Zn doped CuCrO2. Authors can calculate the relation of resistivity, Hall mobility and carrier concentrations. Furthermore, the Hall mobility should be compared with conventional p-type semiconductors. If the calculated mobility seems not reliable, authors should discuss only the resistivity in the paper.

4.The process temperature (annealing) of 600 dC is rather high for practical use. For flat-panel displays, 400 dC is the maximum available temperature for mass-production. Regarding this point, authors should comment how to decrease the process temperature or examples from already reported sol-gel process oxide films.

Author Response

Thank you for the reviewers’ positive comments. I have carefully read the comments and revised the manuscript according to reviewers’ suggestions. All the changes of technical content are marked blue font in the revised manuscript. A brief description of my response to each point raised by the reviewers is as follows:

#Reviewer 2

In this paper, authors have successfully fabricated Zn doped CuCrO2 thin films for p-type TCO using solution-process. This paper is scientifically interesting for a variety of applications for optoelectronics devices. Therefore, this paper should be published in this journal. However, before this work is publishable, the authors should respond to the following concerns.

1.I agree with the importance of developing p-type TCOs. I suggest several papers to enhance the motivation of this work: J. Kim et al., Advanced Materials Interfaces, 5(23), 1801307., T. Jun et al., Advanced Materials 30 (12), 1706573.

It will be better that authors can explain the detailed necessity of p-type TCOs in the introduction part of this paper. Work functions of conventional n-type TCOs and their problems can be discussed.  

The previous literatures have reported the importance of developing p-type transparent/oxide semiconductors and its applicability to device electronics [27,28]. The work functions of conventional n/p- type TCOs and their problems and solutions also have been discussed comprehensively. We all have the same purposes, which is conductive to the practical optoelectronic applications for these semiconductors.   

27.   J. Kim; K. Yamamoto; S. Iimura; S. Ueda; H. Hosono, Electron affinity control of amorphous oxide semiconductors and its applicability to organic electronics, Adv. Mater. Interfaces 2018, 5, 1801307, doi: 10.1002/admi.201801307

28.   T. Jun; J. Kim;  M. Sasase;  H. Hosono, Material design of p-type transparent amorphous semiconductor, Cu–Sn–I, Adv. Mater. 2018, 30, 1706573, doi: 10.1002/adma.201706573

2. If possible, authors should comment on the work function of the CuCrO2.

Fabrication of such a p-n heterojunction device makes it possible to develop the p-type CuCr1−xZnxO2 applications, such as selective contacts in photovoltaics, and p-channel transparent film transistors.

3.Even if the mobility is low, authors must indicates the Hall mobility of Zn doped CuCrO2. Authors can calculate the relation of resistivity, Hall mobility and carrier concentrations. Furthermore, the Hall mobility should be compared with conventional p-type semiconductors. If the calculated mobility seems not reliable, authors should discuss only the resistivity in the paper.

The CuCr1−xZnxO2 (x = 0.065, 0.085) films with lower resistivities were 3.82 and 3.98 Ωcm, respectively, and the carrier mobility were 8.65 and 9.38 cm−3, respectively. The values are close to conventional p-type semiconductors. General speaking, when compared to n-type semiconductors, p-type materials have lower mobility due to a hole current can only be evolved when electron holes are filled with valence electrons in the neighborhood, not free electrons.  

4.The process temperature (annealing) of 600 dC is rather high for practical use. For flat-panel displays, 400 dC is the maximum available temperature for mass-production. Regarding this point, authors should comment how to decrease the process temperature or examples from already reported sol-gel process oxide films.

To achieve the available temperature of the mass-production below 400 °C. The high energy generated by excimer laser is incident on the surface of the CuCrO2 film, and a thermal energy effect is generated only at a depth of 100 nm on the film surface. Therefore, too much thermal energy is not transferred to the substrate or material underneath. This annealing method is benefit for CuCrO2 development in the future.

Round  2

Reviewer 1 Report

The authors have complied with all points raised by this reviewer. The paper is improved and ready for publication.